# Identification of Estrogen Signaling in a Prioritization Study of Intraocular Pressure-Associated Genes

**DOI:** 10.3390/ijms221910288

**Published:** 2021-09-24

**Authors:** Hannah A. Youngblood, Emily Parker, Jingwen Cai, Kristin Perkumas, Hongfang Yu, Jason Sun, Sylvia B. Smith, Kathryn E. Bollinger, Janey L. Wiggs, Louis R. Pasquale, Michael A. Hauser, W. Daniel Stamer, Yutao Liu

**Affiliations:** 1Department of Cellular Biology and Anatomy, Augusta University, Augusta, GA 30912, USA; hyoungblood@augusta.edu (H.A.Y.); emparker1@augusta.edu (E.P.); jincai@augusta.edu (J.C.); hyu@augusta.edu (H.Y.); jasonus90@gmail.com (J.S.); sbsmith@augusta.edu (S.B.S.); 2Department of Ophthalmology, Duke University Medical Center, Durham, NC 27710, USA; kristin.perkumas@dm.duke.edu; 3Department of Ophthalmology, Augusta University, Augusta, GA 30912, USA; kbollinger@augusta.edu; 4James & Jean Culver Vision Discovery Institute, Augusta University, Augusta, GA 30912, USA; 5Department of Ophthalmology, Massachusetts Eye and Ear Infirmary, Boston, MA 02114, USA; janey_wiggs@meei.harvard.edu; 6Department of Ophthalmology, Icahn School of Medicine at Mount Sinai, New York, NY 10029, USA; louis.pasquale@mssm.edu; 7Department of Medicine, Duke University, Durham, NC 27710, USA; mike.hauser@duke.edu; 8Department of Ophthalmology and Biomedical Engineering, Duke University, Durham, NC 22710, USA; william.stamer@duke.edu; 9Center for Biotechnology and Genomic Medicine, Augusta University, Augusta, GA 30912, USA

**Keywords:** primary open-angle glaucoma, intraocular pressure, aqueous humor outflow, estrogen signaling

## Abstract

Elevated intraocular pressure (IOP) is the only modifiable risk factor for primary open-angle glaucoma (POAG). Herein we sought to prioritize a set of previously identified IOP-associated genes using novel and previously published datasets. We identified several genes for future study, including several involved in cytoskeletal/extracellular matrix reorganization, cell adhesion, angiogenesis, and TGF-β signaling. Our differential correlation analysis of IOP-associated genes identified 295 pairs of 201 genes with differential correlation. Pathway analysis identified β-estradiol as the top upstream regulator of these genes with *ESR1* mediating 25 interactions. Several genes (i.e., *EFEMP1*, *FOXC1*, and *SPTBN1*) regulated by β-estradiol/*ESR1* were highly expressed in non-glaucomatous human trabecular meshwork (TM) or Schlemm’s canal (SC) cells and specifically expressed in TM/SC cell clusters defined by single-cell RNA-sequencing. We confirmed *ESR1* gene and protein expression in human TM cells and TM/SC tissue with quantitative real-time PCR and immunofluorescence, respectively. 17β-estradiol was identified in bovine, porcine, and human aqueous humor (AH) using ELISA. In conclusion, we have identified estrogen receptor signaling as a key modulator of several IOP-associated genes. The expression of ESR1 and these IOP-associated genes in TM/SC tissue and the presence of 17β-estradiol in AH supports a role for estrogen signaling in IOP regulation.

## 1. Introduction

Glaucoma is the leading cause of irreversible blindness in the world, impacting the lives of approximately 75 million individuals [1,2,3]. Primary open-angle glaucoma (POAG) is the most predominant form of this group of optic neuropathies, accounting for approximately 75% of cases [1,2,3]. Like other forms of glaucoma, POAG is characterized by retinal ganglion cell death and optic nerve atrophy resulting in progressive visual field loss, but differs from other forms of glaucoma in that it lacks an identifiable cause [3,4,5,6,7]. 

Although its etiology is not completely understood, several risk factors have been identified, including age, African ancestry, positive family history, and elevated intraocular pressure (IOP) [1,3,5,7,8,9,10]. IOP is sustained by the dynamic production and outflow of aqueous humor (AH), the fluid which carries nutrients to and removes wastes from the avascular cornea, lens, and trabecular meshwork (TM) in the anterior segment of the eye [3,7,11]. After being produced by the ciliary body (CB), AH flows forward between the lens and iris to the anterior segment where it provides nutrients to the cornea [3,7]. The fluid then flows out of the eye, carrying with it metabolic waste from the lens, iris, and cornea [11]. AH outflow predominantly occurs through the conventional pathway in the iridocorneal angle [12,13]. In this pathway, AH flows through the TM, Schlemm’s canal (SC), and out into the vasculature through the episcleral veins and collector channels [3,7,12]. The tissues in this region, predominantly the TM, regulate outflow resistance [7,13,14,15]. If the outflow resistance of the AH is elevated, the equilibrium of production and outflow is disrupted, and IOP increases [7,13,14,15]. Although elevated IOP is not experienced in all POAG cases, it is the only modifiable risk factor for POAG that slows disease progression, and as such has been the subject of intense investigation [7,10]. 

In order to identify genetic factors that contribute to IOP regulation and POAG risk, many genome-wide association studies (GWAS) have been conducted in recent years [16,17,18,19,20,21,22,23,24,25,26]. Two GWAS conducted in 2018 by Khawaja, et al., and MacGregor, et al., identified single nucleotide polymorphisms (SNPs) in or around more than 150 genes [24,25]. Many of these were novel associations with IOP. Due to limited time and resources, not all of these genes can be thoroughly examined using in vitro and in vivo studies. Therefore, this study sought to prioritize these genes by using previously published data, publicly available databases, and our own unpublished data as screening criteria.

## 2. Results

### 2.1. Selection of Genes for Prioritization

By combining the data provided in the supplemental files from Khawaja, et al., and MacGregor, et al., we compiled a list of 191 lead SNPs associated with IOP (Appendix A). These SNPs were located in or near 157 genes. In some cases, multiple SNPs were identified within a single gene (e.g., *TMCO1*, *FNDC3B*, *AFAP1*, *GMDS*, *OXR1*, *ANGPT1*, *LMX1B*, and *GAS7*). Alternatively, some SNPs were located between genes; in such cases, genes on either side of the SNP were included. Furthermore, many genes were identified by both studies. For this reason, the final list of IOP-associated genes included only 157 non-redundant genes (Appendix A). 

### 2.2. Gene Prioritization

The list of genes were prioritized using multiple criteria (Figure 1). First, differential gene expression was examined in two previously published microarray datasets, one comparing expression in healthy and POAG-affected human TM tissue [27] and the other comparing expression in human SC cells isolated from healthy and POAG-affected donor eyes [28]. The genes identified as being significantly differentially expressed in these datasets were then analyzed for their level of expression in an unpublished RNA-sequencing (RNA-Seq) study we conducted previously in primary cultures of human TM and SC cells. Expression of these genes of interest was also examined in TM, CB, and cornea tissue [29,30]. The genes were further characterized using the Genotype-Tissue Expression (GTEx) Portal [31,32,33,34] and the UCSC Genome Browser (https://genome.ucsc.edu/, accessed on 3 December 2018) [35]. Separately, all 157 IOP-associated genes were screened using expression data from an RNA-Seq study, in which we subjected cultures of primary human TM cells to cyclic mechanical stretch, and a single-cell RNA-sequencing (scRNA-Seq) analysis of human outflow tissues [36,37]. Lastly, we conducted a differential expression correlation analysis of all 157 IOP-associated genes and the genes differentially expressed in POAG-affected human TM tissues [27]. Differential correlation analysis was followed by network analysis.

#### 2.2.1. Differential Expression in POAG-Affected Human TM Tissue or SC Cells

While some of the currently available drug therapies for glaucoma manage IOP by reducing secretion, the primary cause of elevated IOP is reduced outflow facility [7,13,14,15]. The TM is responsible for regulating AH outflow [7,13,14,15]; therefore, understanding the transcriptional differences between healthy and glaucomatous TM is crucial. Genes differentially expressed in the TM of POAG patients may play a role in disease etiology by affecting AH outflow, and therefore IOP. For this reason, the differential expression of the 157 IOP-associated genes was looked up in a previously published microarray comparing POAG-affected (*n* = 14) and non-glaucomatous (*n* = 13) postmortem human TM tissues [27]. From this microarray, 10 differentially expressed (│FC│ ≥ 1.5) genes were identified: *ANKH*, *ANTXR1*, *COL8A2*, *EFEMP1*, *GMDS*, *HLA-DQA1*, *LTBP2*, *MAFB*, *RPLP2*, and *SEMA3E* (Table 1). For these genes, there was evidence for upregulation in POAG-affected TM tissue.

As AH exits the TM, it enters the SC lumen, a vessel that has both vascular and lymphatic characteristics [12,38]. The endothelial cells comprising the inner wall of the SC share responsibility with the TM for outflow resistance and IOP regulation [12,13,14,15,38,39]. As a result, genes differentially expressed in the SC cells of POAG patients are likely involved in POAG pathogenesis. Therefore, the differential expression of the 157 IOP-associated genes was cross-referenced in previously published microarray data comparing gene expression in glaucomatous (*n* = 4) and non-glaucomatous (*n* = 4) SC cells [28]. There were eight genes that were upregulated (i.e., *AFAP1*, *COL4A1*, *CPXM1*, *ETS1*, *PRKAG2*, *PTPN1*, *SH2B3*, and *TES*), and nine that were downregulated (i.e., *ANGPTL2*, *ANTXR1*, *HHEX*, *LTBP2*, *MAFB*, *ME3*, *NR1H3*, *PCSK5*, and *PRSS23*) in POAG-affected SC cells (Table 2). Three of these genes (i.e., *ANTXR1*, *LTBP2*, and *MAFB*) had also been shown to be differentially expressed in POAG-affected human TM tissue (Figure 2, Table 1). Interestingly, the direction of differential expression for the 3 shared genes was opposite for TM tissue and SC cells.

#### 2.2.2. High Expression in RNA-Seq of Primary Human TM and SC Cells

In order to examine the expression of the 24 differentially expressed IOP-associated genes in normal tissue, we conducted stranded total RNA-Seq of primary TM and SC cells from post-mortem donors with no history of ocular illness. While the 10 TM genes in Table 1 were filtered using RNA-Seq expression data from normal TM cells, the 17 SC genes in Table 2 were analyzed using normal SC cell RNA-Seq expression data. Genes having an average number of counts (FPKM) ≥ 10 were considered to be highly expressed. These genes included 6 differentially expressed genes in POAG-affected TM tissue and 11 differentially expressed genes in POAG-affected SC cells. Two of these genes (i.e., *ANTXR1* and *LTBP2*) had been differentially expressed in both POAG-affected TM tissue and SC cells, resulting in 15 non-redundant IOP-associated genes having high expression in normal TM and/or SC cells (Figure 2, Table 3). 

#### 2.2.3. High Expression in Human Ocular Tissue 

To confirm gene expression in normal tissue, we examined the expression of these 24 genes of interest in TM, CB, and corneal tissue in the Human Ocular Tissue Database and a previously published transcriptomic profile of these same tissues [29]. As mentioned previously, the CB and TM are the tissues responsible for producing and draining AH, respectively. In addition, several studies have shown a relationship between glaucoma risk and central corneal thickness (CCT), particularly a thin CCT [40,41]. Because of the association with disease risk, we also examined corneal tissue. 

Four of these 24 genes (i.e., *ANKH*, *EFEMP1*, *PTPN1*, and *TES*) were found to be significantly expressed (DABG ≤ 0.05) in TM tissue from the Human Ocular Tissue Database [29] and highly expressed (FPKM ≥ 10) in TM tissue profiled by Carnes, et al. [30] (Figure 2, Table 4 and Table 5). Furthermore, both studies found *ANKH*, *EFEMP1*, and *TES* to be expressed in either CB or cornea tissue (Table 4 and Table 5). Interestingly, all genes that were highly expressed in TM cells were highly expressed in at least one of the normal TM tissue profiles. On the other hand, *SEMA3E*, *HLA-DQA1*, and *MAFB*, which had low or no expression in normal TM cells (Table 3), were highly expressed in at least one of the normal TM tissue profiles (Table 4 and Table 5), suggesting that two-dimensional TM cell culture does not completely capture the biological processes that occur in tissue.

#### 2.2.4. Variant Impacts on Expression Due to Location in Regulatory Regions

Having identified 15 genes that were differentially expressed between pathogenic and healthy outflow cells/tissue and specifically expressed at high levels in non-glaucomatous tissue, we sought to identify how the IOP-associated SNPs in/near these genes might be affecting their expression. Gene expression may be regulated by a variety of mechanisms. One such way is the binding of transcription factors at promoter, enhancer, or repressor regions upstream or downstream of the transcription start site. A single nucleotide change in one of these regions could be sufficient to impact transcription factor recognition of and/or binding affinity to the site. In addition to genetic regulatory regions, epigenetic modification of histones can cause transcriptional changes by increasing or decreasing chromatin condensation, thereby influencing the exposure of gene promoters to transcriptional machinery. Just as a single nucleotide change in a regulatory region could impact transcription initiation, a SNP located in a region of histone modification could affect the ability of the histone to be modified, thereby influencing chromatin condensation and transcription. Therefore, we identified the location of each of the IOP-associated SNPs in/near the genes of interest using the UCSC Genome Browser [35] (http://genome.ucsc.edu/, accessed on 3 December 2018). All of the 15 SNPs were located in regions of histone modification, and 8 were located in promoter/enhancer regions (Figure 2, Table 6). 

In order to better elucidate how these SNPs might affect gene expression, we sought to identify tissues in which these gene/SNP pairs might form expression quantitative trait loci (eQTL) using the publicly available GTEx Portal (https://gtexportal.org, accessed on 3 December 2018) [31,32,33,34]. Although eye tissue had not been included in the GTEx Project, identification of significant eQTL for these gene/SNP pairs could provide evidentiary support for SNP effects on gene expression. In addition, identifying other tissues in which these SNPs significantly impact gene expression (i.e., significant eQTL) might elucidate how the gene/SNP pair functionally impact outflow tissue.

Significant eQTL were identified for 10 of the 15 gene/SNP pairs in one or more tissues with skin, tibial nerve, and whole blood being the tissues with the most eQTL (Table 6). The SNP rs55892100 had significantly affected *TES* gene expression in more tissues than any other gene/SNP pair, suggesting that this SNP might be in a region universally important for gene expression. 

#### 2.2.5. Differential Expression in Cyclically Stretched Primary Human TM Cells

Due to their pressure-responsive regulation of AH outflow, TM cells are under constant mechanical stretch [7,13,14,15,42,43]. Furthermore, the heart beat and filling of the choroid creates an ocular pulse of AH outflow through the TM, creating a cyclical process of mechanical stretch [15,43,44,45,46]. In order to identify IOP-associated genes responsible for regulating TM cell response to this stress, we analyzed differential expression of all 157 IOP-associated genes in our previously published RNA-Seq dataset of primary human TM cells subjected to cyclic mechanical stretch [36]. 

Four IOP-associated genes were significantly differentially expressed (│FC│ > 2, *p* < 0.05) between stretched and unstretched TM cells (Table 7). These included *MCPH1*, *MAFB*, *HGF*, and *PCSK5*. Both *MAFB* and *PCSK5* had been differentially expressed in POAG-affected SC cells (Figure 2, Table 2). In both cases, the genes were downregulated. In addition, 4 other IOP-associated genes (i.e., *MYOF*, *FOXC1*, *SPTBN1*, and *TCF7L2*) shared sequence homology to long non-coding RNAs (lncRNAs) differentially expressed (│FC│ > 2, *p* < 0.05) in cyclically stretched human TM cells (Table 8), suggesting that these IOP-associated genes may be regulated by lncRNAs within the TM in response to stretch. 

#### 2.2.6. Differential Expression in Outflow Pathway scRNA-Seq

The transcriptional profile of cell types within the AH outflow pathway has been recently characterized by scRNA-Seq [37,47]. As a last confirmation of the expression of IOP-associated genes within the outflow pathway, we analyzed which of the 157 IOP-associated genes were specifically expressed in cell clusters defined by the publicly available scRNA-Seq dataset provided by van Zyl, et al. [37]. Due to the nature of sequencing RNA transcripts from single cells, coverage is often low, even given sufficiently large read depths. As a result, scRNA-Seq mostly detects genes highly expressed in the tissue of interest, to the exclusion of some lower expressed genes. Therefore, even the transcripts that are detected at a relatively low abundance in a scRNA-Seq dataset would have to be highly expressed in the native tissue in order to be detected. For this reason, we did not apply an expression cutoff for inclusion, but rather included any IOP-associated genes that were specifically expressed within one or more scRNA-Seq cell clusters.

Twenty-nine IOP-associated genes were specifically expressed in one of the 19 cell clusters (Figure 3, Appendix A). Because these 19 clusters included some clusters which described cell types that are not specific to the outflow pathway, such as melanocytes, neurons, and pericytes, we further restricted our analysis to cell clusters specifically involved in AH production and outflow. These 6 cell clusters were localized to the TM (i.e., Beam Cell A, Beam Cell B, and Cribiform Juxtacanalicular (JCT)), SC (i.e., SC Endothelium), distal outflow pathway (i.e., Collector Channel/Aqueous Vein), and CB (i.e., Ciliary Muscle). Twenty-one genes were specifically expressed in one or more of these clusters, with *RPLP2* being expressed in all six cell clusters and having the highest expression. *SPTBN1*, *FOXC1*, and *EMCN* were the genes most commonly expressed in the TM and SC. Eight of the 21 genes (i.e., *ANKH*, *EFEMP1*, *GMDS*, *HLA-DQA1*, *MAFB*, *PRSS23*, *RPLP2*, and *TES*) had been differentially expressed in POAG-affected TM tissue or SC cells (Figure 2, Table 1 and Table 2). Interestingly, *MAFB* had also been differentially expressed in cyclically stretched human TM cells (Figure 2, Table 8). Furthermore, the expression of *TES* in this scRNA-Seq study further supports evidence for its importance identified by prior filtering criteria (Figure 2). 

Of note, van Zyl, et al. found that the three TM cell clusters highly, but differentially, expressed one of our genes of interest *EFEMP1*, with the JCT cluster having the highest expression and the Beam Cell A cluster having the lowest expression [37]. It is important to note the differences between the TM cell clusters: the TM Beam Cell A cluster preferentially expressed *FABP4* while the TM Beam Cell B cluster preferentially expressed *TMEFF2* and was localized closer to the JCT [37]. Therefore, it is possible that the cell A cluster corresponded with the uveal TM while the cell B cluster corresponded to the corneoscleral TM.

#### 2.2.7. Differential Expression Correlation Analysis 

Lastly, in order to identify potential interactions between IOP-associated genes, we performed a differential correlation analysis using previously published expression data from control and POAG human TM tissue [27]. All differentially expressed genes in this dataset (i.e., 480 genes) were also included in the differential correlation analysis along with the 157 IOP-associated genes. Genes were evaluated for expression correlations within groups (i.e., control or POAG), that is gene-gene pairs that followed similar expression patterns. These expression correlations would suggest that the two genes might affect each other’s expression or be a part of a common cascade. Then it was determined whether there was a significant difference between control and POAG expression correlations for these gene-gene pairs. Differential correlations would suggest that one or both of the genes in the correlation pair, or the biological network they share in common, may be involved in POAG pathogenesis. 

From this differential correlation analysis, we identified 295 significant differential correlation pairs (FDR ≤ 0.1) comprised of 201 genes (Appendix A). Of these 201 genes, 45 were IOP-associated genes. To better understand how these correlation pairs interacted, we created an interaction network for all correlation pairs using Cytoscape (Figure 4). Many of the 201 genes were involved in more than one correlation pair, with *GCNT3*, *MUC20*, *KLK11*, *PLK5*, and *IL1RN* being involved in the most number of correlation pairs.

In order to examine known functional relationships between these genes and better understand what biological processes they may impact, we conducted a network analysis of all 201 genes using Ingenuity Pathway Analysis (IPA). β-estradiol was identified as the top upstream regulator of these genes, with *ESR1* (i.e., estrogen receptor 1 or α) mediating regulatory interactions with 25 genes (Figure 5). Twelve of these genes were associated with IOP (i.e., *ANGPTL2*, *CAV2*, *DLL1*, *EFEMP1*, *FOXC1*, *IGF1*, *SCAMP1*, *SEMA3F*, *SOS2*, *SPTBN1*, *TCF72*, and *VEGFC*). In addition, network analysis identified three networks centered on β-estradiol and estrogen receptors *ESR2* (i.e., estrogen receptor 2 or β) and *GPER1* (i.e., G protein-coupled estrogen receptor 1) (Appendix A). This suggests that these IOP-associated genes may have a significant role in the regulation of IOP and AH outflow and may be modulated by estrogen signaling.

Dexamethasone, a corticosteroid known to increase AH outflow resistance and IOP, was also identified as one of the top upstream regulators of the 201 genes identified by differential correlation analysis (Figure 5) [48,49,50,51]. This ocular hypertension-inducing molecule regulated 44 of the 201 genes, with 9 of these genes being associated with IOP (i.e., *ADAM12*, *CAV2*, *DGKG*, *DLL1*, *FANCA*, *FBXO32*, *IGF1*, *SPTBN1*, and *VEGFC*). Several of these genes, including *CAV2*, *DLL1*, *IGF1*, *SPTBN1*, and *VEGFC*, were also regulated by *ESR1*, suggesting shared, but possibly opposing, mechanisms of IOP regulation. 

### 2.3. Estrogen Signaling in the Human Outflow Pathway

The expression of estrogen receptors, including *ESR1*, has been established in multiple ocular tissues including cornea, iris, lens, CB, retina, and conjunctiva [52]. Furthermore, *ESR1* has been shown to be expressed in human TM cells [53]. Moreover, the various isoforms of estrogen have been shown to be present in bovine, canine, rabbit, and human AH [54,55,56,57] as well as in human TM cells [53]. 

Here we sought to confirm receptor expression in human outflow tissue and the presence of 17β-estradiol, the active isoform of estrogen, in human AH. We confirmed the presence of 17β-estradiol in both glaucomatous (34.46 ± 7.08 pg/mL, *n* = 4) and non-glaucomatous (37.90 ± 14.80 pg/mL, *n* = 4) human AH, as well as bovine (59.70 ± 8.58 pg/mL, *n* = 5) and porcine (149.44 ± 168.21 pg/mL, *n* = 4) AH (Table 9; Appendix A). Unfortunately, there were insufficient numbers of male and female human AH samples per group to statistically analyze sex differences between human AH estradiol concentrations, and the biological sex for bovine and porcine AH samples was unavailable. However, there were no trending differences between human male and female AH samples. 

The expression of estrogen receptors *ESR1*, *ESR2*, and *GPER1* was ascertained in human TM cells (Appendix A) by quantitative real-time PCR (qRT-PCR). *ESR1* was expressed at detectable levels in females (*n* = 3, Ct = 29.37), but not in males (*n* = 3, Ct = 35.16). Interestingly, *ESR2* was expressed at barely detectable levels in males (Ct = 33.90), but was undetectable in females (Ct = 34.69). However, *GPER1* was expressed at detectable, equivalent levels in both males and females (Ct = 31.73, 31.04). As confirmation, ESR1 protein was expressed throughout the TM/SC region of outflow tissue derived from human organ donors of both sexes and a range of ages (*n* = 4; Figure 6, Appendix A). Although quantification of ESR1 fluorescence was not conducted, there were no apparent differences between the sexes. Identification of both receptor and ligand in human outflow tissue supports a possible role of *ESR1* signaling in IOP regulation. 

## 3. Discussion

### 3.1. Overview

Herein, we prioritized a set of 157 genes associated with IOP. From this set of genes, we identified several genes of interest based on their high expression in conventional outflow pathway tissue/cells and their differential expression in glaucomatous outflow tissue/cells. Furthermore, this study identified β-estradiol as a key regulator of many IOP-associated genes. ESR1 expression in TM/SC tissue and 17β-estradiol presence in AH supported this potential regulatory role for estrogen signaling.

### 3.2. IOP-Associated Gene Prioritization

Of the 157 genes, several genes passed multiple levels of filtering criteria (Figure 2). Of note, *TES* passed the most levels of prioritization criteria (Figure 2), having been differentially expressed in POAG-affected SC cells (Table 2), being one of the most highly expressed IOP-associated genes in non-glaucomatous SC cells (Table 3), and the highest expressed IOP-associated gene in TM tissue as indicated by the Human Ocular Tissue Database (Table 4). *TES* was also highly expressed in TM tissue assayed by Carnes, et al., (Table 5) as well as specifically expressed in the Beam Cell B and JCT cell clusters defined by scRNA-Seq (Figure 3, Appendix A). The IOP-associated SNP in/near *TES* was located in a region of histone modification and an enhancer/promoter region (Table 6). Moreover, the SNP rs55892100 had significantly affected *TES* gene expression in more tissues than any other gene/SNP pair, suggesting that the SNP must be in a region universally important for gene expression. Furthermore, the association of the *TES* SNP with IOP was quite strong (*p* = 1.5 × 10^−21^; Appendix A), and although its association with POAG was not significant (*p* = 5.3 × 10^−6^) at the level of genome-wide significance, it was suggestive [25]. Furthermore, the recent GWAS conducted by Gharahkhani, et al., identified a SNP between *TES* and *TFEC* that was significantly associated (*p* = 1.06 × 10^−9^) with IOP in a European discovery cohort after Conditional and Joint (COJO) analysis [26]. However, significance for this SNP was not confirmed in the study’s cross-ancestry meta-analysis or in the replication cohort [26], suggesting that its association is limited to individuals of European ancestry. 

*TES* (i.e., testin) is a tumor suppressor gene expressed in multiple tissues, where it is involved in the formation of focal adhesions and actin stress fibers [58,59,60]. In fact, Sala, et al., has shown that testin is a mechanosensitive component of the actin cytoskeleton and that it localizes to actin stress fibers in response to activated RhoA [61]. Focal adhesions and actin cytoskeletal elements play important roles in TM cell contractility and response to mechanical strain [49,62,63], and Rho/ROCK signaling has been shown to decrease AH outflow facility by inducing TM cell contraction [64,65]. Therefore, given the multiple lines of evidence supporting its significance in the outflow pathway, *TES* and its IOP-associated SNP may be of special interest for future functional studies. 

In addition to *TES*, several other genes passed multiple levels of filtering criteria including *ANTXR1*, *ETS1*, and *HHEX*. *ANTXR1* was differentially expressed (│FC│ ≥ 1.5) in POAG-affected TM and SC cells (Figure 2, Table 1 and Table 2) while *ETS1* and *HHEX* were differentially expressed in POAG-affected SC cells alone (Figure 2, Table 2). Moreover, *ANTXR1* and *ETS1* were highly expressed (FPKM ≥ 10) in non-glaucomatous TM and/or SC cells (Figure 2, Table 3) and highly expressed (FPKM ≥ 10) in the TM, CB, or cornea tissue profiled by Carnes, et al. (Table 5) [30]. Additionally, IOP-associated variants in/near *ANTXR1* and *ETS1* were located in either an enhancer or promoter region or a region of histone modification (Figure 2, Table 6). Both *ETS1* and *HHEX* act as transcription factors [66,67]. Interestingly, *HHEX* inhibits the repression of WNT signaling, a signaling pathway involved in normal IOP regulation and dysregulated in glaucoma [68,69,70]. Like *TES*, *ANTXR1* (i.e., anthrax toxin receptor 1) is involved in mediating actin cytoskeleton adhesion to substrate [71].

Many other prioritized genes (e.g., *AFAP1*, *ANTXR1*, *COL4A1*, *COL8A2*, *EFEMP1*, *ETS1*, *LTBP2*, *PCSK5*, *PTPN1*, and *SEMA3E*) also play roles in cytoskeletal reorganization, extracellular matrix (ECM) assembly, and/or cell adhesion, especially through focal adhesions [71,72,73,74,75,76,77,78,79,80,81,82,83,84]. It has been well established that ECM composition, cytoskeletal organization, and cell adhesion are critical to TM regulation of IOP homeostasis [13,14,15,49,85,86,87,88]. Furthermore, glaucomatous TM is characterized by dysregulation of these processes. For example, glaucomatous TM demonstrates an abnormal ECM content and an altered actin cytoskeleton [15]. Furthermore, cytoskeletal reorganization, ECM turnover, and focal adhesion assembly are processes responsive to mechanical strain of the TM [62,89,90,91]. Interestingly, *ETS1* (i.e., ETS proto-oncogene 1, transcription factor) is responsive to both fluid shear [92,93,94] and mechanical stress [95,96], types of stress that are regularly experienced by TM cells [15,89,90], and *AFAP1* is necessary for c-Src-mediated mechanotransduction [77]. Perhaps these genes mediate IOP homeostasis through ECM modulation and cytoskeletal reorganization in response to mechanical strain. 

However, IOP-associated genes may affect IOP regulation through several other processes. For example, another key biological process shared by several prioritized genes (e.g., *ANGPTL2*, *ANTXR1*, *COL4A1*, *COL8A2*, *ETS1*, and *SEMA3E*) was angiogenesis, blood vessel morphogenesis, and endothelial cell regulation [97,98,99,100,101,102,103,104,105,106,107]. Perhaps these genes are involved in IOP regulation through the role of the SC endothelium and the distal vasculature in aqueous humor outflow [12,13,14,15,38,39,108,109,110,111]. Therefore, genes prioritized by our analysis may affect IOP homeostasis by several mechanisms.

### 3.3. Estrogen Signaling in IOP Regulation

Our network analysis identified β-estradiol as being a key upstream regulator of a number of IOP-associated genes with many of these regulatory interactions being coordinated by *ESR1*. It should be noted that the genes regulated by estradiol were not only associated with IOP, but some also passed several filtering criteria, indicating a high likelihood that these estrogen-regulated genes play important roles in outflow regulation. The presence of estradiol in AH and the expression of estrogen receptors, including *ESR1*, in human TM/SC tissue further suggest a possible role for estrogen in IOP regulation.

Over the past couple decades, there has been increasing evidence that estrogen signaling plays a role in IOP homeostasis and protecting against glaucoma. For example, it has been shown that IOP levels and risk for developing glaucoma increase at menopause [112]. Conversely, post-menopausal hormone replacement therapy, including estrogen-only treatment, has been shown to decrease IOP and glaucoma risk [112,113,114,115,116,117,118,119]. Similarly, shorter lifetime estrogen exposure due to late menarche or early menopause has been shown to increase risk for glaucoma [120]. Furthermore, the effects of estrogen levels on IOP are not limited to post-menopausal individuals. IOP has also been shown to vary during the menstrual cycle with IOP being lower during the hyperestrogenic luteal phase [121]. Furthermore, IOP decreases in an estrogen-level dependent manner during pregnancy, especially during the second trimester, an elevated estrogen state [122,123,124]. Moreover, this decrease in IOP appears to result from an increased outflow facility [122]. Interestingly, antiestrogen therapies, such as tamoxifen, have been shown to exert detrimental ocular effects, including elevated IOP [125]. However, it must be noted that several studies have found the opposite trend or have found no significant difference in IOP or glaucoma risk/incidence [126].

In addition to the epidemiological evidence, Kosior-Jareck, et al., have demonstrated that POAG phenotypes may be affected by *ESR1* variants [127]. Furthermore, *CYP1B1*, which is responsible for metabolizing 17β-estradiol, is associated with POAG as well as primary congenital glaucoma [128]. In fact, an entire panel of SNPs within genes in the estrogen metabolic pathway was shown to be associated with glaucoma, including high-tension glaucoma [129].

Beyond the clinical and genetic studies, a few functional studies have also implicated estrogen signaling in the regulation of IOP homeostasis. Chen, et al., observed an 8–19% IOP increase in female aromatase knockout mice, which lack estradiol synthesis [130]. Several studies have validated that *ESR1* and other estrogen receptors are expressed throughout the eye [116,131,132]. The various forms of estrogen, including the active form 17β-estradiol (0.83 ng/mL), have been identified previously in bovine AH [54]. Furthermore, the presence of estradiol in human TM cells has been confirmed [53], as was the expression of several estrogen metabolic enzymes, including aromatase [53]. Mookherjee, et al., have shown that estradiol treatment of TM cells upregulates *MYOC* through estrogen regulatory elements, and that decreasing estradiol metabolism through *CYP1B1* mutagenesis increases *MYOC* expression [53]. They postulate that this connection between *CYP1B1* and *MYOC* may explain the digenic mode of inheritance seen in some cases of juvenile glaucoma [53]. A separate study has shown that estradiol treatment of human TM cells results in increased nitric oxide production, which could lead to vasodilation downstream [133]. Furthermore, 17β-estradiol has been suggested to exert protective effects in retinal tissue via sigma-1 receptor, agonism of which has been shown to lower IOP in both normo- and hypertensive rodent models [134,135]. 

However, despite the plethora of data evidencing that estrogen signaling plays an important role in IOP homeostasis, the mechanism of this effect has not been fully explored. Here we present 12 genes (i.e., *ANGPTL2*, *CAV2*, *DLL1*, *EFEMP1*, *FOXC1*, *IGF1*, *SCAMP1*, *SEMA3F*, *SOS2*, *SPTBN1*, *TCF72*, and *VEGFC*) that are both regulated by estrogen and are associated with IOP. Perhaps these genes and the common biological processes they share can illuminate our understanding of estrogen’s impact on IOP dynamics. 

The majority of these estrogen-regulated, IOP-associated genes (i.e., *ANGPTL2*, *CAV2*, *DLL1*, *FOXC1*, *IGF1*, *TCF7L2*, and *VEGFC*) are involved in angiogenesis, endothelial cell function, or other aspects of the vasculature [100,136,137,138,139,140,141,142,143,144]. *FOXC1* and *VEGFC* are also involved in lymphangiogenesis [139,145]. These genes may transduce estrogen-mediated effects on IOP through the distal vessels and the SC, given the combined lymphatic and vascular nature of its endothelial cells [38]. Furthermore, estrogen is known to stimulate vasodilation through nitric oxide production, a known regulator of AH outflow [133,146]. 

As seen with the prioritized genes discussed previously, other common biological processes shared by estrogen-regulated genes (i.e., *CAV2*, *DLL1*, *EFEMP1*, *SEMA3F*, and *SPTBN1*) include cytoskeletal reorganization, ECM turnover, and cell adhesion [78,84,147,148,149,150,151,152,153,154]. Interestingly, *CAV2* has been shown to be essential for remodeling TM ECM [151]. Of note, several SNPs in *CAV1/2* have been identified as being more strongly associated in women [155], further supporting a role of estrogen regulation of *CAV2*.

Furthermore, three of these genes are involved in ocular development and/or pathology. *DLL1* is required for normal eye development [156] while *EFEMP1* has been implicated in age-related macular degeneration and is also associated with Malattia Leventinese and Doyne honeycomb retinal dystrophy [157,158]. Of note, *FOXC1* is required for proper development of the anterior segment and has been previously associated with congenital glaucoma [159].

To further elucidate how estrogen signaling may contribute to IOP homeostasis, it may be helpful to note that several of the genes (i.e., *CAV2*, *DLL1*, *IGF1*, *SPTBN1*, and *VEGFC*) regulated by estradiol/*ESR1* were also regulated by dexamethasone, suggesting that the two molecules may affect similar processes. Dexamethasone is known to elevate IOP by increasing TM cell and ECM stiffness, thereby increasing outflow resistance [48,49,50,51]. This further supports the possibility that estrogen signaling may impact IOP homeostasis through the TM ECM. 

Taken together, these findings suggest that estrogen signaling may contribute to maintaining IOP homeostasis through endothelial function in the SC/distal vasculature and through homeostatic modulation of the TM cytoskeleton/ECM in response to IOP fluctuations. Estrogen signaling may also contribute to IOP homeostasis through ocular development or modulation of glaucoma-related pathways. 

### 3.4. Limitations and Future Work

Despite the rigor of consulting multiple data sources, a few limitations must be noted for this study. First, genes were selected based on their proximity to the SNPs identified by GWAS. However, the SNPs in question may occur in regulatory elements for genes much further downstream or upstream. Second, a few of the datasets were limited by small sample sizes. Our own RNA-Seq sample sets of non-glaucomatous TM (*n* = 4) and SC cells (*n* = 2) and stretched TM cells (*n* = 5) [36] were small as were the TM, ciliary body, and cornea tissue sample sets (*n* = 4) used for RNA-Seq by Carnes, et al. [30]. In addition, although their sample sizes were sufficiently large, the scRNA-Seq dataset referenced by this study identified only ~24,000 single-cell transcriptomes, thereby limiting the power of their results. The third limitation of this study is that TM tissue dissection was likely performed differently between studies. Therefore, different cell types may be included in the different tissue profiles. This could account for discrepancies in gene expression between TM tissue datasets. Fourth, differences between two-dimensional cell culture and tissue may account for differences between *ESR1* expression in male human TM cells and male human TM/SC tissue. Fifth, due to the small number of samples, human AH estradiol concentrations were not able to be statistically analyzed on the basis of sex. Furthermore, the biological sex of bovine and porcine AH was unknown although it was assumed to be male due to commonly accepted abattoir practices. Despite these limitations, including multiple datasets with different approaches and both cells and tissue provides rigor for our prioritization study and ensures that genes of interest have several sources of support.

In the future, additional transcriptional studies with larger sample sizes and more standardized approaches may be consulted to further support our findings. In addition, functional assays may be conducted to determine whether associated SNPs impact IOP regulation through our genes of interest and whether these effects on IOP have pathological consequences. Furthermore, significant effort should be directed towards elucidating the mechanism by which estrogen signaling plays a role in IOP regulation.

### 3.5. Conclusions

In summary, we have prioritized a set of 157 genes containing/neighboring IOP-associated SNPs. As a result, we have identified several genes of interest for future functional study as well as identified estradiol as a key regulator of several IOP-associated genes. Therefore, we postulate a role for estrogen signaling in the normal maintenance of IOP homeostasis. Although much evidence for estrogenic effects on IOP regulation has accumulated over the years, the mechanism of action has not been well studied. Here we present a group of genes that are both associated with IOP and regulated by estrogen, thereby offering insight into the mechanisms by which estrogen may impact IOP. Given the function of the estrogen-regulated, IOP-associated genes, we hypothesize that estrogen may mediate its effects on IOP through dilation of the SC/distal vessels and/or through modulation of TM ECM and cellular morphology. The confirmed expression of ESR1 in TM/SC tissue and the presence of estradiol in AH further supports this hypothesis.

## 4. Materials and Methods

### 4.1. Gene Prioritization Using RNA-Sequencing and Previously Published Datasets

Appendix A from the GWAS by Khawaja, et al., and MacGregor, et al., were used to identify genes of interest [24,25]. In their Appendix A, Khawaja, et al., identified 133 SNPs associated with IOP; these SNPs were located in or near 142 genes (Appendix A). Meanwhile, MacGregor, et al., identified 106 IOP-associated SNPs/genes in their Appendix A (Appendix A). The 133 SNPs from Khawaja, et al., were combined with the 106 SNPs identified by MacGregor, et al., to yield a total of 191 non-redundant IOP-associated SNPs in or near 157 non-redundant genes (Appendix A). 

First, the list of 157 IOP-associated genes (Appendix A) was screened using microarray data from a study previously published by our group [27]. In brief, human TM tissue was obtained from 14 POAG patients during trabeculectomy and from 13 nonglaucomatous post-mortem eyes. Differential gene expression between POAG and control TM tissue was determined using a HumanWG-6 BeadChip array (Illumina, San Diego, CA, USA). For our current study, genes were considered to be differentially expressed if │FC│ ≥ 1.5, regardless of *p*-value (Table 1). 

Second, differential expression of the 157 IOP-associated genes (Appendix A) was examined in our previously published microarray dataset of POAG-affected human SC cells [28]. In brief, SC cells were isolated from 4 glaucomatous and 4 non-glaucomatous post-mortem human donor eyes. Differential gene expression was analyzed using a HumanWG-6 BeadChip array (Illumina, San Diego, CA, USA). A │FC│ ≥ 1.5 was the sole criterion for differential expression in our current study (Table 2). 

A series of additional filtering criteria were then applied to the 24 genes that were differentially expressed in human TM tissue or SC cells. First, expression of the 10 genes differentially expressed in POAG-affected TM tissue (Table 1) was determined in four non-glaucomatous primary human TM cell lines using total stranded RNA-Seq (Table 3). Likewise, expression of the 17 genes differentially expressed in POAG-affected SC cells (Table 2) was determined in 2 non-glaucomatous primary human SC cell lines using total stranded RNA-Seq (Table 3). Genes with an average expression (FPKM) ≥ 10 were considered to be highly expressed (Table 3). Second, the expression for each of the 24 genes in the Human Ocular Tissue Database (https://genome.uiowa.edu/otdb/, accessed on 12 December 2020; ©2019 The Center for Bioinformatics & Computational Biology at The University of Iowa) [29] was determined (Table 4). Search criteria was restricted to trabecular meshwork, CB, and corneal tissues; values from the core; and values from reverse strand probes only. Third, we determined the expression (FPKM) of the 24 genes in a previously published transcriptional profile of adult human TM, CB, and cornea tissue [30]. In brief, the North Carolina Eye Bank provided tissue samples from four adult post-mortem human donors. Resulting RNA libraries were sequenced by Illumina Hi-Seq (Illumina, San Diego, CA, USA). Genes with an average expression (FPKM) ≥ 10 were considered to be highly expressed (Table 5). Fifth, the University of California, Santa Cruz (UCSC) Genome Browser (http://genome.ucsc.edu/cgi-bin/hgGateway, accessed on 3 December 2018; ©2000–2018 The Regents of The University of California) [35] was employed to determine whether the 15 differentially and highly expressed IOP-associated SNPs were located in enhancer or promoter regions or in regions of histone modification, using the GRCh37/hg19 human genome assembly (Table 6). Lastly, the Genotype-Tissue Expression (GTEx) Portal (https://gtexportal.org/, accessed on 3 December 2018; ©2021 Broad Institute of MIT and Harvard; V6P release with 544 donors, GENCODE version v19, GRCh37/hg19) [31,32,33,34] was used to identify in which tissues the 15 differentially and highly expressed IOP-associated gene-SNP pairs formed eQTL (Table 6). 

Next, all 157 genes associated with IOP (Appendix A) were screened against a previously published RNA-Seq dataset of normal TM cells subjected to cyclic mechanical stretch [36]. In brief, primary cultures of human TM cells were obtained from 5 donors with no history of ocular disease. Secondary cultures of these cells were subjected to 24 hours cyclic mechanical stretch (15% stretch, 1 cycle/second) with the computer-controlled FX-5000 Tension System (Flexcell International Corporation, Burlington, NC, USA). Stranded total RNA-sequencing was conducted using a NextSeq 500 System (Illumina, San Diego, CA, USA). For this study, mRNAs and lncRNAs with │FC│ > 2 and *p* < 0.05 were considered to be significantly differentially expressed (Table 7 and Table 8). 

Lastly, the specific expression of all 157 IOP-associated genes (Appendix A) was examined in a previously published scRNA-Seq of human outflow tissue [37]. In brief, whole eyes were obtained from 7 post-mortem human donors, and a “permissive dissection technique” was used to remove TM and surrounding tissue from the anterior segment. Single cell suspensions were sequenced using 10× Chromium Single Cell Chips (10× Genomics, Pleasanton, CA, USA) and Illumina HiSeq 2500. The median expression for each of the 157 genes was determined using the Single Cell Portal (https://singlecell.broadinstitute.org/single_cell/study/SCP780/cell-atlas-of-aqueous-humor-outflow-pathways-in-eyes-of-humans-and-four-model-species-provides-insights-into-glaucoma-pathogenesis, accessed on 8 March 2021) [37] and applying default settings (Appendix A). 

### 4.2. Differential Expression Correlation and Network Analyses

Microarray expression data [27] was used to determine the expression of the 157 IOP-associated genes in both glaucomatous (POAG) and non-glaucomatous TM tissue. In addition, 480 genes were differentially expressed between glaucomatous and non-glaucomatous TM tissue samples in this dataset. The expression levels of the 157 IOP genes and the 480 differentially expressed TM genes were imported into RStudio (version 3.5.3, RStudio, Boston, MA, USA) and differential correlation analysis based upon this expression data was conducted as described previously [160]. In brief, after removing duplicated genes and genes with no expression, gene-gene expression correlations were identified within groups (i.e., glaucomatous or non-glaucomatous tissue) using R’s Pearson correlation function (cor). The R psych package’s test for significant correlation (r.test) was used to calculate the significance (*P*) of differences between gene-gene correlations in glaucomatous and non-glaucomatous TM tissue. In this way, gene-gene correlations identified in non-glaucomatous tissue that changed significantly (e.g., no correlation or oppositely signed correlation) in glaucomatous tissue could be identified, and vice versa. R’s Benjamini-Hochberg procedure function (*p*.adjust with “BH” parameter) was then used to calculate the FDR from the *p*-value. Duplicate correlations were removed, and gene-gene correlations with FDR ≤ 0.01 were considered significant and retained for further analysis.

All significantly correlated genes were input into Cytoscape (version 3.5.1, Cytoscape Consortium, New York, NY, USA) [161] in order to visualize interactions between gene-gene correlation pairs. Furthermore, IPA (version 65367011, Qiagen, Hilden, Germany) was used to identify biological interactions and functional networks involving these significantly correlated genes.

### 4.3. Reagents

For RNA-sequencing, mirVana miRNA Isolation kits (catalog number AM1560) were obtained from Thermo Fisher Scientific (Waltham, MA, USA), High Sensitivity DNA kits (catalog number 5067–4626) from Agilent Technologies (Santa Clara, CA, USA), and both RiboGone Mammalian kits (catalog number 634847) and SMARTer Stranded Total RNA Low Input Mammalian Sample Prep kits (catalog number 634861) from TaKaRa Bio USA, Inc. (San Jose, CA, USA). RNeasy Mini kits (catalog number 74104) were purchased from Qiagen (Hilden, Germany) and High-Capacity cDNA Reverse Transcription kits with RNase Inhibitor (catalog number 4374966) were purchased from Applied Biosystems (Waltham, MA, USA). 17β-estradiol ELISA kits (catalog number ADI-900–174) were obtained from Enzo Life Sciences (Farmingdale, NY, USA). All primers were obtained from Integrated DNA Technologies (Coralville, IA, USA; Appendix A). For immunofluorescence, citrate buffer (catalog number 21545) was purchased from Millipore Sigma (Burlington, MA, USA), Triton X-100 (catalog number 1610407) from Bio-Rad Laboratories, Inc. (Hercules, CA, USA), bovine serum albumin (BSA; catalog number a8022) from Sigma-Aldrich (St. Louis, MO, USA), and ProLong^TM^ Diamond Antifade Mountant with DAPI (catalog number 36966) from Invitrogen (Waltham, MA, USA). Mouse anti-ESR1 (catalog number MA1–310) and Cy^TM^3 goat anti-mouse IgG (catalog number 115–165-062) antibodies were purchased from Invitrogen and Jackson ImmunoResearch Laboratories, Inc. (West Grove, PA, USA), respectively (Appendix A). 

### 4.4. RNA-Sequencing

Four primary TM cell lines and two primary SC cell lines were derived from post-mortem human cornea rims obtained from Duke University. Cultures were derived, maintained, and characterized according to accepted standards [162,163]. RNA isolation and stranded total RNA-sequencing were conducted as described previously [160]. In brief, a mirVana miRNA Isolation kit (Thermo Fisher Scientific Waltham, MA, USA) was used to extract total RNA from the 6 cell lines. RNA quality was assayed with a High Sensitivity DNA kit using Agilent 2100 Bioanalyzer (Agilent Technologies, Santa Clara, CA, USA), with an RNA Integrity Number (RIN) ≥ 6 being required for sequencing. Ribosomal RNA (rRNA) was depleted with a RiboGone Mammalian kit (TaKaRa Bio USA, Inc., San Jose, CA, USA). A SMARTer Stranded RNA-Seq kit (TaKaRa Bio USA, Inc., San Jose, CA, USA) was used to generate cDNA libraries from 200 ng total RNA. Pooled cDNA libraries underwent 50bp paired-end sequencing on an Illumina HiSeq 2500 system (Integrated Genomics Shared Resource, Georgia Cancer Center, Augusta University, Augusta, GA, USA). Alignment against human Ensembl transcriptome or lncRNAs annotated from NONCODE was performed by TopHat 2.1.1 [164] following quality control and demultiplexing, and Cufflinks software determined read counts in fragments per kilobase per million reads (FPKM). An average coverage of at least 25–30 million reads was achieved for each sample. 

### 4.5. Aqueous Humor Estradiol Quantification

Glaucomatous (*n* = 4) and nonglaucomatous (*n* = 4) AH samples were obtained from the Eye Clinic at Augusta University Medical Center. The study was approved by the Institutional Review Board at Augusta University. Written consent was collected from all individuals. Samples were collected from male and female patients undergoing cataract or glaucoma surgery (Appendix A). Bovine (*n* = 5) and porcine (*n* = 4) AH were obtained from the local abattoir. After performing a liquid-liquid extraction of 17β-estradiol with 5:1 (*v/v*) diethyl ether:sample, the presence of 17β-estradiol in these samples was confirmed with enzyme-linked immunosorbent assay (ELISA; Enzo Life Sciences, Farmingdale, NY, USA) following the manufacturer’s protocol. Optical density was determined at 405nm using an Infinite M200 Pro plate reader (Tecan, Männedorf, Switzerland).

### 4.6. Quantitative Real-Time PCR

Primary human TM cell lines (*n* = 6; Appendix A) were derived from post-mortem human cornea rims obtained from the Eye Surgery Center of Augusta and the Department of Ophthalmology at Augusta University (Augusta, GA, USA). Cell lines were cultured and characterized according to accepted standards [162,163]. RNA was extracted using a Qiagen RNeasy Mini kit following the manufacturer’s protocol. RNA quality was assayed with a High Sensitivity DNA kit and 2100 Bioanalyzer (Agilent Technologies, Santa Clara, CA, USA), with an RIN ≥ 6 being required for inclusion. cDNA was reverse transcribed from 100ng RNA using a High-Capacity cDNA Reverse Transcription kit (Applied Biosystems, Waltham, MA, USA) following the manufacturer’s protocol. A 10μL reaction was prepared using iTaq Universal SYBR Green Supermix (Bio-Rad Laboratories, Inc., Hercules, CA, USA) and 10μM of the following primers: β-actin, *ESR1*, *ESR2*, and *GPER1* (Appendix A). β-actin was used as an endogenous control. The reactions were run on a StepOnePlus System (Applied Biosystems, Waltham, MA, USA) and relative expression levels were calculated using the ΔΔC_t_ method, with female human TM cells being considered the control sample.

### 4.7. Immunofluorescence 

Discarded cornea rims (*n* = 4; Appendix A) were obtained from the Eye Surgery Center of Augusta and the Department of Ophthalmology at Augusta University (Augusta, GA, USA) following cornea transplant. The rims were hemisected and a ~500 mm wedge removed for histology. The corneal rim section was placed in Davidson’s fixative for >24 h, before being moved to 70% EtOH. The corneal rim section was then paraffin-embedded and sectioned by the Augusta University Electron Microscopy and Histology Core. Three sections per sample were processed for H&E staining. Six sections (i.e., 3 sections for negative staining and 3 sections for positive staining) were prepared for immunofluorescence by xylene de-paraffinization, ethanol rehydration, and citrate buffer (1×, Millipore Sigma, Burlington, MA, USA) antigen retrieval. Tissue membranes were permeabilized with Triton X-100 (Bio-Rad Laboratories, Inc., Hercules, CA, USA) before blocking in 5% BSA (Sigma-Aldrich, St. Louis, MO, USA) for 1 hour. Slides were then incubated with 1:100 mouse anti-ESR1 antibody (Invitrogen, Waltham, MA, USA) overnight at 4 °C before incubating with 1:100 secondary antibody (Cy^TM^3 goat anti-mouse IgG; Jackson ImmunoResearch Laboratories, Inc., West Grove, PA, USA) for 30 minutes of darkness at room temperature. ProLong^TM^ Diamond Antifade Mountant (Invitrogen, Waltham, MA, USA) was used to fix coverslips before imaging with an Axio Imager D.2 microscope (Zeiss, Oberkochen, Germany) and Zeiss Zen 2.3 pro software (version 2.3, Zeiss, Oberkochen, Germany).

### 4.8. Statistical Analysis

The filtering analyses described in Section 4.1 and the statistical analysis for the aqueous humor quantification described in Section 4.5 were performed using Microsoft Excel 2016 (version Office 16, Microsoft Corporation, Redmond, WA, USA) as were ΔΔCt calculations for the qRT-PCR described in Section 4.6. The differential correlation analysis described in Section 4.2 was performed using RStudio (version 3.5.3, RStudio, Boston, MA, USA).

## Figures and Tables

**Figure 1 ijms-22-10288-f001:**
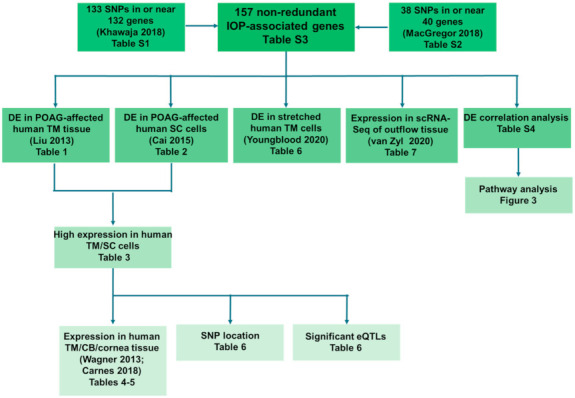
Experimental design for prioritization of intraocular pressure (IOP) associated genes. From two previously published genome-wide association studies (GWAS) [24,25], we identified 157 non-redundant IOP-associated genes, which we filtered and characterized for future functional analysis using the following criteria: (1) differential expression (DE) in primary open-angle glaucoma (POAG) affected trabecular meshwork (TM) tissue, (2) DE in POAG-affected Schlemm’s canal (SC) cells, (3) high expression in non-glaucomatous human TM and/or SC cells, (4) significant/high expression in human TM, ciliary body (CB), and cornea tissue, (5) single nucleotide polymorphism (SNP) location in regulatory regions, (6) significant expression quantitative trait loci (eQTL) for gene/SNP pairs, (7) DE in cyclically stretched human TM cells, (8) specific expression in cell clusters defined by single-cell RNA-sequencing (scRNA-Seq) of human conventional outflow tissue, and (9) pathway analysis of differential correlation pairs.

**Figure 2 ijms-22-10288-f002:**
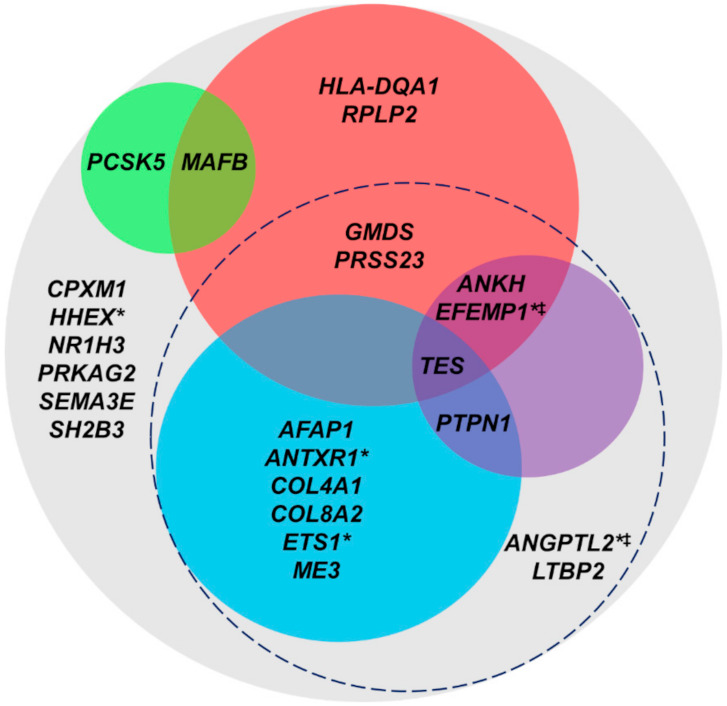
Prioritization of intraocular pressure (IOP)-associated genes differentially expressed in primary open-angle glaucoma (POAG)-affected trabecular meshwork (TM) tissue and/or Schlemm’s canal (SC) cells. Twenty-four IOP-associated genes were differentially expressed (│FC│ ≥ 1.5) in POAG-affected TM tissue and/or POAG-affected SC cells (grey circle). Fifteen of the 24 genes were highly expressed (FPKM ≥ 10) in non-glaucomatous TM or SC cells (dashed circle), all of which were located in regions of histone modification. Eight of the IOP-associated variants in/near these 15 genes were located in enhancer or promoter regions (blue circle), and 4 of the 15 were significantly/highly expressed in TM tissue (purple circle). Eight of the 24 genes were specifically expressed in ≥1 cell cluster defined by single-cell RNA-sequencing of human outflow tissue (red circle), and 2 of the 24 genes were differentially expressed by cyclically stretched human TM cells (green circle). Five of the 24 genes were part of differential correlation pairs (asterisk) and 2 were regulated by *ESR1* (two-barred cross).

**Figure 3 ijms-22-10288-f003:**
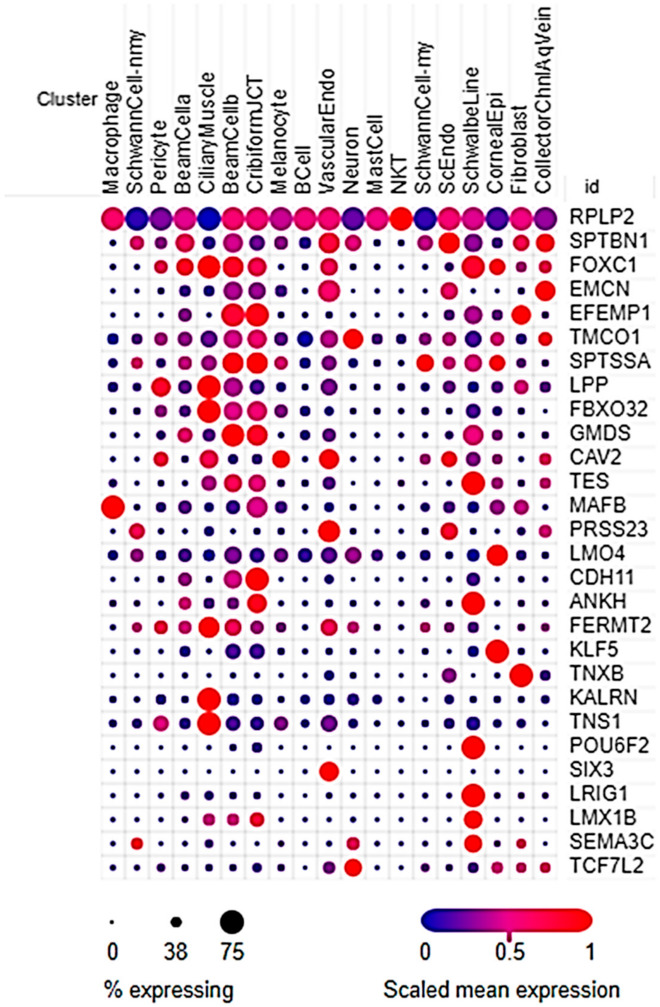
Genes expressed in single-cell RNA-sequencing (scRNA-Seq) cell clusters. Twenty-nine intraocular pressure-associated genes were specifically expressed in one or more of the 19 cell clusters defined by scRNA-Seq of human aqueous humor outflow tissue. Genes expressed in a high percentage of cells within a cluster are represented by a large dot while genes expressed in a low percentage of cells are represented by a small dot. Highly expressed genes appear more red while lowly expressed genes appear more blue. Cell clusters included Macrophage, Non-Myelinated Schwann Cell (SchwannCell-nmy), Pericyte, Trabecular Meshwork Beam Cell A, Ciliary Muscle, Trabecular Meshwork Beam Cell B, Cribiform Juxtacanalicular Trabecular Meshwork Cell (CribiformJCT), Melanocyte, B Cell, Vascular Endothelium (VascularEndo), Neuron, Mast Cell, NKT Cell, Myelinated Schwann Cell (SchwannCell-my), Schlemm’s Canal Endothelium (ScEndo), Schwalbe Line Cell, Corneal Epithelium (CornealEpi), Fibroblast, and Collector Channel/Aqueous Vein Endothelium (CollectorChnlAqVein).

**Figure 4 ijms-22-10288-f004:**
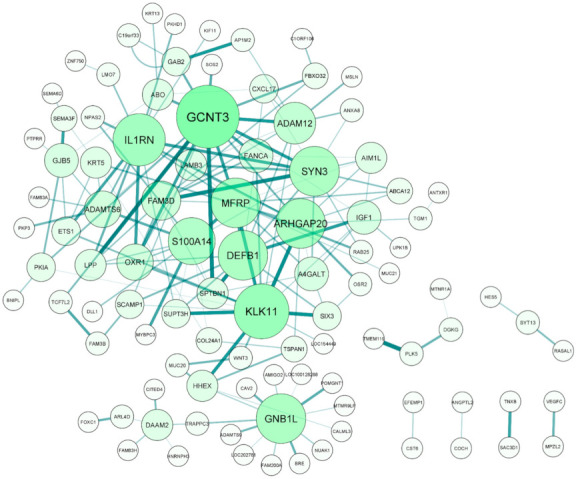
Complex interactions of differential correlation pairs involving intraocular pressure (IOP)-associated genes. Differential correlation analysis identified 295 differential correlation pairs (FDR ≤ 0.1) comprised of 201 genes. Of these, 166 pairs included 45 IOP-associated genes. Due to many IOP-associated genes being a part of multiple differential correlation pairs, these gene pairs formed several complex correlation networks. The size and color of the nodes indicate which genes are involved in the most differential correlation pairs with the larger green nodes having the most interactions. Similarly, the size and color of the lines connecting the gene nodes indicate the significance of the differential correlation with thicker, darker lines representing greater significance.

**Figure 5 ijms-22-10288-f005:**
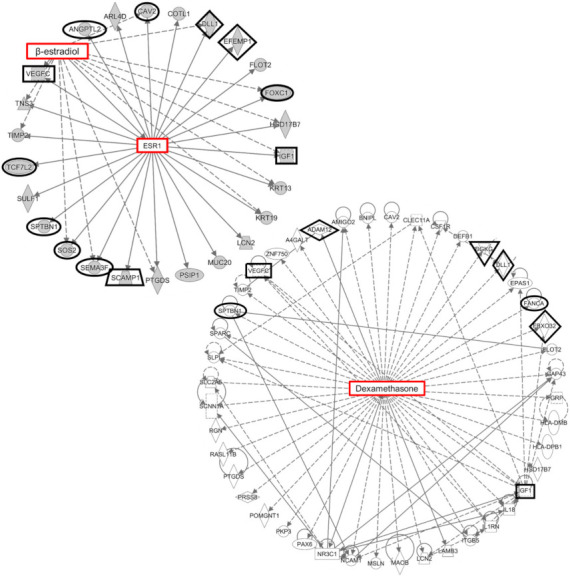
Functional networks composed of genes identified by differential correlational analysis. The 201 genes identified by differential correlation analysis comprised several functional networks. The top upstream regulator of the 201 genes identified by differential correlation analysis was β-estradiol with *ESR1* mediating regulatory interactions between β-estradiol and 25 genes. Another top-upstream regulator was dexamethasone, which regulated several of the same genes as β-estradiol. Intraocular pressure-associated genes are indicated by a thick black outline.

**Figure 6 ijms-22-10288-f006:**
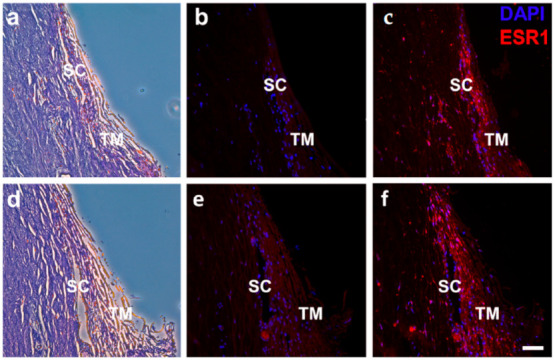
Expression of ESR1 in the trabecular meshwork (TM)/Sclemm’s canal (SC) region of human outflow tissue. ESR1 protein was expressed in the TM/SC region of human outflow tissue (*n* = 4, 24–60 years of age) taken from both male (**a**–**c**) and female (**d**–**f**) post-mortem donors. Representative images of H&E (**a**,**d**), negative control (**b**,**e**), and positive staining (**c**,**f**). ESR1, estrogen receptor 1; TM, trabecular meshwork; SC, Schlemm’s canal. Scale bar: 50 μm.

**Table 1 ijms-22-10288-t001:** Intraocular Pressure-Associated Genes Differentially Expressed in Primary Open-Angle Glaucoma-Affected Human Trabecular Meshwork Tissue.

Gene	Description	Fold Change	*p*
*ANKH*	ANKH inorganic pyrophosphate transport regulator	2.69	3.16 × 10^-7^
*ANTXR1*	ANTXR cell adhesion molecule 1	1.79	6.46 × 10^-3^
*COL8A2*	Collagen type VIII α2 chain	2.66	1.32 × 10^-4^
*EFEMP1*	EGF containing fibulin extracellular matrix protein 1	1.82	4.43 × 10^-3^
*GMDS*	GDP-mannose 4,6-dehydratase	1.93	3.14 × 10^-3^
*HLA-DQA1*	Major histocompatibility complex, class II, DQ α1	2.13	3.10 × 10^-3^
*LTBP2*	Latent TGF-β binding protein 2	2.25	1.65 × 10^-3^
*MAFB*	MAF BZIP transcription factor B	1.54	1.22 × 10^-2^
*RPLP2*	Ribosomal protein lateral stalk subunit P2	1.58	1.32 × 10^-1^
*SEMA3E*	Semaphorin 3E	1.94	1.26 × 10^-5^

**Table 2 ijms-22-10288-t002:** Intraocular Pressure-Associated Genes Differentially Expressed in Primary Open-Angle Glaucoma-Affected Human Schlemm’s Canal Cells.

Gene	Description	Fold Change	*p*
*AFAP1*	Actin filament associated protein 1	1.56	1.35 × 10^−1^
*ANGPTL2*	Angiopoietin like 2	−3.33	6.73 × 10^−2^
*ANTXR1*	ANTXR cell adhesion molecule 1	−1.67	4.77 × 10^−2^
*COL4A1*	Collagen type IV α1 chain	2.84	1.43 × 10^−1^
*CPXM1*	Carboxypeptidase X, M14 family member 1	1.57	2.18 × 10^−2^
*ETS1*	ETS proto-oncogene 1, transcription factor	1.61	1.28 × 10^−1^
*HHEX*	Hematopoietically expressed homeobox	−1.62	1.81 × 10^−2^
*LTBP2*	Latent TGF-β binding protein 2	−1.63	1.56 × 10^−1^
*MAFB*	MAF BZIP transcription factor B	−1.79	2.82 × 10^−1^
*ME3*	Malic enzyme 3	−1.59	3.23 × 10^−3^
*NR1H3*	Nuclear receptor subfamily 1 group H member 3	−1.59	5.54 × 10^−2^
*PCSK5*	Proprotein convertase subtilisin/kexin type 5	−2.04	3.93 × 10^−2^
*PRKAG2*	Protein kinase AMP-activated non-catalytic subunit γ2	1.63	1.70 × 10^−2^
*PRSS23*	Serine protease 23	−1.65	1.60 × 10^−1^
*PTPN1*	Protein tyrosine phosphatase non-receptor type 1	1.63	1.63 × 10^−2^
*SH2B3*	SH2B adaptor protein 3	1.58	7.07 × 10^−3^
*TES*	Testin LIM domain protein	1.54	4.75 × 10^−2^

**Table 3 ijms-22-10288-t003:** Expression of Differentially Expressed Intraocular Pressure-Associated Genes in Non-Glaucomatous Human Trabecular Meshwork/Schlemm’s Canal Cells

Gene	Cell/Tissue Type	RNA-Seq Expression (FPKM)
*AFAP1*	SC	46.64
*ANGPTL2*	SC	30.90
*ANKH*	TM	64.41
*ANTXR1 **	TM, SC	32.78, 47.13
*COL4A1*	SC	17.92
*COL8A2*	TM	34.44
*CPXM1*	SC	0.49
*EFEMP1*	TM	89.70
*ETS1*	SC	15.70
*GMDS*	TM	29.40
*HHEX*	SC	2.60
*HLA-DQA1*	TM	0.00
*LTBP2 **	TM, SC	104.70, 171.74
*MAFB **	TM, SC	1.37, 1.79
*ME3*	SC	11.09
*NR1H3*	SC	4.45
*PCSK5*	SC	1.97
*PRKAG2*	SC	6.49
*PRSS23*	SC	184.92
*PTPN1*	SC	12.46
*RPLP2P1*	TM	0.00
*SEMA3E*	TM	1.51
*SH2B3*	SC	13.63
*TES*	SC	64.48

* Differentially expressed in both trabecular meshwork tissue and Schlemm’s canal cells.

**Table 4 ijms-22-10288-t004:** Expression of Differentially Expressed Intraocular Pressure-Associated Genes in the Human Ocular Tissue Database.

Gene	Trabecular Meshwork			Ciliary Body			Cornea		
	PLIER	DABG Average	DABG Range	PLIER	DABG Average	DABG Range	PLIER	DABG Average	DABG Range
*AFAP1*	33.64	0.08	0–0.58	31.53	0.07	0–0.49	30.54	0.14	0–0.77
*ANGPTL2*	45.11	0.16	0–0.71	30.32	0.19	0–0.8	74.34	0.07	0–0.42
*ANKH*	112.61	0.05	0–0.76	70.15	0.05	0–0.51	88.56	0.07	0–0.51
*ANTXR1 **	145.19	0.06	0–0.89	90.84	0.10	0–0.77	58.56	0.09	0–0.84
*COL4A1*	46.71	0.08	0–0.92	42.61	0.16	0–0.97	24.59	0.24	0–0.98
*COL8A2*	99.82	0.12	0–0.56	86.39	0.14	0–0.52	81.40	0.11	0–0.57
*CPXM1*	26.48	0.11	0–0.58	24.60	0.17	0–0.85	26.29	0.10	0–0.32
*EFEMP1*	214.28	0.01	0–0.11	667.78	0.03	0–0.35	165.29	0.04	0–0.52
*ETS1*	86.82	0.07	0–0.76	41.16	0.08	0–0.51	27.07	0.05	0–0.35
*GMDS*	76.23	0.06	0–0.20	88.91	0.19	0–0.78	93.19	0.06	0–0.16
*HHEX*	92.41	0.14	0–0.64	63.12	0.20	0–0.54	95.97	0.23	0–0.95
*HLA-DQA1*	NA	NA	NA	NA	NA	NA	NA	NA	NA
*LTBP2 **	79.44	0.06	0–0.66	69.94	0.08	0–0.77	62.10	0.08	0–0.53
*MAFB **	54.38	0.19	0–0.93	62.34	0.17	0–0.82	77.52	0.09	0–0.50
*ME3*	29.17	0.13	0–0.86	36.24	0.17	0–0.52	34.63	0.11	0–0.52
*NR1H3*	32.84	0.18	0–0.72	32.88	0.23	0–1.00	33.11	0.16	0–0.57
*PCSK5*	37.87	0.07	0–0.33	31.31	0.17	0–0.81	234.57	0.01	0–0.18
*PRKAG2*	65.93	0.08	0–0.68	57.75	0.14	0–0.78	51.41	0.13	0–0.72
*PRSS23*	76.78	0.06	0–0.34	155.24	0.02	0–0.15	121.46	0.04	0–0.28
*PTPN1*	84.10	0.02	0–0.22	61.15	0.06	0–0.62	78.01	0.07	0–0.37
*RPLP2P1*	NA	NA	NA	NA	NA	NA	NA	NA	NA
*SEMA3E*	130.67	0.01	0–0.20	28.40	0.10	0–0.82	35.46	0.07	0–0.83
*SH2B3*	40.00	0.15	0–0.99	29.77	0.15	0–0.68	37.72	0.12	0–0.5
*TES*	290.39	0.01	0–0.11	67.06	0.07	0–0.27	193.21	0.01	0–0.08

* Differentially expressed in both trabecular meshwork tissue and Schlemm’s canal cells.

**Table 5 ijms-22-10288-t005:** Expression (FPKM) of Filtered Intraocular Pressure-Associated Genes in a Previously Published Transcriptional Profile of Adult Human Trabecular Meshwork, Ciliary Body, and Cornea Tissue.

Gene	Trabecular Meshwork	Ciliary Body	Cornea
*AFAP1*	3.10	4.71	2.46
*ANGPTL2*	18.84	15.75	8.69
*ANKH*	50.88	32.83	21.43
*ANTXR1 **	67.92	5.40	10.42
*COL4A1*	9.15	8.06	0.73
*COL8A2*	118.54	33.40	28.26
*CPXM1*	6.63	9.52	0.22
*EFEMP1*	142.27	153.01	77.16
*ETS1*	16.18	1.85	3.56
*GMDS*	81.59	55.42	89.55
*HHEX*	2.25	0.53	0.48
*HLA-DQA1*	31.13	4.11	9.36
*LTBP2 **	54.69	142.61	3.93
*MAFB **	13.35	1.45	8.08
*ME3*	7.26	15.74	5.42
*NR1H3*	8.60	19.88	7.02
*PCSK5*	5.04	2.71	21.57
*PRKAG2*	16.79	2.30	16.08
*PRSS23*	23.22	8.20	16.50
*PTPN1*	16.16	34.22	17.24
*RPLP2P1*	0.00	0.00	0.00
*SEMA3E*	8.27	0.16	1.93
*SH2B3*	4.54	2.79	1.28
*TES*	56.81	0.81	30.17

* Differentially expressed in both trabecular meshwork tissue and Schlemm’s canal cells.

**Table 6 ijms-22-10288-t006:** Genomic Location of Intraocular Pressure (IOP)-Associated Single Nucleotide Polymorphisms (SNPs) and Expression Quantitative Trait Loci (eQTL) for IOP-Associated Gene/SNP Pairs.

SNP	Gene	Histone Modification	Enhancer or Promoter Region	Significant eQTL
rs28649910	*AFAP1*	Yes	Yes	Skin; Whole blood
rs11795066	*ANGPTL2*	Yes	No	Cerebellum; Esophageal mucosa; Skin; Testis; Tibial nerve; Thyroid; Whole blood
rs368503	*ANKH*	Yes	No	Esophageal mucosa
rs6546486	*ANTXR1 **	Yes	Yes	None
rs112972174	*COL4A1*	Yes	Yes	None
rs12123086	*COL8A2*	Yes	Yes	Whole blood
rs4672075	*EFEMP1*	Yes	No	Skin; Thyroid
rs7924522	*ETS1*	Yes	Yes	None
rs722585	*GMDS*	Yes	No	None
rs74384554	*LTBP2 **	Yes	No	Tibial nerve
rs2433414	*ME3*	Yes	Yes	Esophageal muscularis; Lung; Skeletal muscle; Skin; Subcutaneous adipose; Tibial nerve; Transformed fibroblasts
rs11606902	*PRSS23*	Yes	No	Adrenal gland; Esophagus; Skeletal muscle
rs6095946	*PTPN1*	Yes	Yes	None
rs10774624	*SH2B3*	Yes	No	Esophageal mucosa; Skin
rs55892100	*TES*	Yes	Yes	Adrenal gland; Lung; Pancreas; Subcutaneous adipose; Testis; Tibial artery; Tibial nerve; Thyroid; Whole blood

* Differentially expressed in both trabecular meshwork tissue and Schlemm’s canal cells.

**Table 7 ijms-22-10288-t007:** Intraocular Pressure-Associated Genes Differentially Expressed in Cyclically Stretched Human Trabecular Meshwork Cells.

Gene	Description	Fold Change	*p*
MCPH1	Microcephalin 1	2.18	2.4 × 10^−2^
MAFB	MAF BZIP transcription factor B	−2.12	1.9 × 10^−2^
HGF	Hepatocyte growth factor	−2.52	3.1 × 10^−2^
PCSK5	Proprotein convertase subtilisin/kexin type 5	−2.75	3.2 × 10^−2^

**Table 8 ijms-22-10288-t008:** Intraocular Pressure-Associated Genes with Sequence Homology to Differentially Expressed Long Non-Coding RNAs (lncRNAs) in Cyclically Stretched Human Trabecular Meshwork Cells.

lncRNA	Sequence Homology	Fold Change	*p*
NONHSAT157336.1:208–1549	Myoferlin	1.61 × 10^5^	3.6 × 10^−2^
NONHSAT106523.2:1153–1748	Forkhead box C1	2.26 × 10^3^	4.5 × 10^−2^
NONHSAT184643.1:226–631	Spectrin β, non-erythrocytic 1	4.77	1.6 × 10^−2^
NONHSAT156133.1:88–552	Transcription factor 7 like 2	−2.42	2.5 × 10^−2^

**Table 9 ijms-22-10288-t009:** 17β-Estradiol Concentration in Bovine, Porcine, and Human Aqueous Humor.

Donor	17β-Estradiol Concentration	
Non-glaucomatous (*n* = 4)	0.14 ± 0.05 nM	37.90 ± 14.80 pg/mL
Glaucomatous (*n* = 4)	0.13 ± 0.03 nM	34.46 ± 7.08 pg/mL
Bovine (*n* = 5)	0.22 ± 0.03 nM	59.70 ± 8.58 pg/mL
Porcine (*n* = 4)	0.55 ± 0.62 nM	149.44 ± 168.21 pg/mL

## Data Availability

The RNA-Seq data is in the process of being deposited into the NCBI GEO database. This will be updated once the deposit is completed with the GEO accession number.

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
