# Peer review of "Identification of Estrogen Signaling in a Prioritization Study of Intraocular Pressure-Associated Genes"

_ijms, 2021, doi:10.3390/ijms221910288_

Round 1
Reviewer 1 Report
The paper entitled “Identification of Estrogen Signaling in a Prioritization Study of 2
Intraocular Pressure-Associated Genes” by Youngblood et al., deals with the
identification of β-estradiol as a key regulator of many IOP-associated genes.
The paper is well written and addresses an issue of interest in the field, however the authors may wish to consider the following prior to publication.
Because the IJMS is not a specialistic eye journal, I suggest making appealing the paper to most of the readers that are not familiar with the ocular field. On this regard, the authors should mention (maybe on line 478) that: “Incidentally, estrogens protect retinal tissue via sigma-1 receptor, and this latter has key role to regulate IOP (please report the following relevant paper (Eur J Pharmacol. 2004 Sep 13;498(1-3):111-4. doi: 10.1016/j.ejphar.2004.06.0; J. Pharmacol. Exp. Ther., 289 (3), 1362-1369, 1999)”. This is appealing because glaucoma is an optic neuropathy characterized by retinal ganglion cell death, irreversible peripheral and central visual field loss other than increase of IOP.
Author Response
Reviewer 1:
The paper entitled “Identification of Estrogen Signaling in a Prioritization Study of Intraocular Pressure-Associated Genes” by Youngblood et al., deals with the identification of β-estradiol as a key regulator of many IOP-associated genes. The paper is well written and addresses an issue of interest in the field, however the authors may wish to consider the following prior to publication.
Author Response:
Thank you for your kind review. We appreciate your helpful suggestions. The additions you suggested have strengthened our manuscript.
Comment 1:
Because the IJMS is not a specialistic eye journal, I suggest making appealing the paper to most of the readers that are not familiar with the ocular field. On this regard, the authors should mention (maybe on line 478) that: “Incidentally, estrogens protect retinal tissue via sigma-1 receptor, and this latter has key role to regulate IOP (please report the following relevant paper (Eur J Pharmacol. 2004 Sep 13;498(1-3):111-4. doi: 10.1016/j.ejphar.2004.06.0; J. Pharmacol. Exp. Ther., 289 (3), 1362-1369, 1999)”. This is appealing because glaucoma is an optic neuropathy characterized by retinal ganglion cell death, irreversible peripheral and central visual field loss other than increase of IOP.
Author Response:
Thank you for the excellent suggestion. We inserted your suggested citation in another paragraph (lines 520-522) for better coherence and flow as follows: “Furthermore, 17β-estradiol has been suggested to exert protective effects in retinal tissue via sigma-1 receptor, agonism of which has been shown to lower IOP in both normo- and hypertensive rodent models [141, 142].”
Reviewer 2 Report
Youngblood et al. wrote a very interesting review describing the role of “Identification of Estrogen Signaling in a Prioritization Study of Intraocular Pressure-Associated Genes”. The manuscript represents an interesting way to discover new scenarios for retinal degenerations. I suggest only several minor revisions needed to update and improve the reliability of the paper:
- The paper is globally too long, and probably should be reduced, especially in several sections like 4.1.
- The 4.8 “Statistical analysis” section should be more detailed.
- Lines 528-531. The authors should update the bibliography related to pathways shared by estrogen signaling and already known as listed ones. Regarding these, I suggest to add the following references to manuscript PMID: 33801777, PMID: 34058230 and PMID: 34440511.
- Finally, manuscript requires English revisions and typos correction.
Author Response
Reviewer 2:
Youngblood et al. wrote a very interesting review describing the role of “Identification of Estrogen Signaling in a Prioritization Study of Intraocular Pressure-Associated Genes”. The manuscript represents an interesting way to discover new scenarios for retinal degenerations. I suggest only several minor revisions needed to update and improve the reliability of the paper:
Author Response:
Thank you so much for your review. Your comments were very helpful in strengthening the quality of our manuscript.
Comment 1:
The paper is globally too long, and probably should be reduced, especially in several sections like 4.1.
Author Response:
We appreciate this comment. We have eliminated several lines that appeared unnecessary in order to reduce the manuscript length, including lines 185-189, 212, 469-473, 549-554, 581-583, 624-629, 642-644, 656-659, 677-688, and 695.
Comment 2:
The 4.8 “Statistical analysis” section should be more detailed.
Author Response:
Thank you for your suggestion. We have added more detail to the 4.8 “Statistical Analysis” section so that it was revised to read as follows (lines 817-821): “The filtering analyses described in section 4.1 and the statistical analysis for the aqueous humor quantification described in section 4.5 were performed using Microsoft Excel 2016 (Redmond, WA, USA) as were ΔΔCt calculations for the quantitative real-time PCR de-scribed in section 4.6. The differential correlation analysis described in section 4.2 was performed using RStudio Version 3.5.3.”
Comment 3:
Lines 528-531. The authors should update the bibliography related to pathways shared by estrogen signaling and already known as listed ones. Regarding these, I suggest to add the following references to manuscript PMID: 33801777, PMID: 34058230 and PMID: 34440511.
Author Response:
Thank you for the suggestion. We have updated the references accordingly.
Comment 4:
Finally, manuscript requires English revisions and typos correction.
Author Response:
Thank you for your careful revision. We have reviewed the manuscript for grammar and spelling and revised accordingly.
Reviewer 3 Report
In the current study, the authors performed extensive amount of analyses in trying to detect intraocular pressure (IOP)-associated genes. The results after network analyses suggested that ESR1 expression in TM and/or SC and the subsequent estrogen receptor signaling might be key players in regulation of IOP. The study was well organized and the manuscript was well written. As a reviewer, I have raised just a point to scientifically improve the manuscript.
Although the previous paper (ref. 28) had already reported using small number of patients, is there gender difference in the amount of β-estradiol in the aqueous humor and/or the expression of ESR1 in TM and/or SC in humans? Because estrogen is a sex hormone, this point should be discussed in the text.
Author Response
Reviewer 3:
In the current study, the authors performed extensive amount of analyses in trying to detect intraocular pressure (IOP)-associated genes. The results after network analyses suggested that ESR1 expression in TM and/or SC and the subsequent estrogen receptor signaling might be key players in regulation of IOP. The study was well organized and the manuscript was well written. As a reviewer, I have raised just a point to scientifically improve the manuscript.
Author Response:
Thank you for the time spent reviewing our manuscript. We appreciate your careful review and helpful suggestions. The corrections you suggested have significantly improved the rigor of our study and the quality of our manuscript.
Comment 1:
Although the previous paper (ref. 28) had already reported using small number of patients, is there gender difference in the amount of β-estradiol in the aqueous humor and/or the expression of ESR1 in TM and/or SC in humans? Because estrogen is a sex hormone, this point should be discussed in the text.
Author Response:
Thank you for this excellent critique. Unfortunately, our human AH estradiol quantifications and immunofluorescent ESR1 protein assays did not have enough male and female samples per group to examine statistical differences between the sexes. However, there appeared to be no trending differences between the sexes. Furthermore, the biological sex data for the bovine and porcine AH was unavailable, but is thought to be male due to common abattoir practices. We have revised the manuscript in several places to reflect this weakness as shown below:
Lines 372-375: “Unfortunately, there were insufficient numbers of male and female human AH samples per group to statistically analyze sex differences between human AH estradiol concentrations, and the biological sex for bovine and porcine AH samples was unavailable. How-ever, there were no trending differences between human male and female AH samples.”
Lines 383-385: “Although quantification of ESR1 fluorescence was not conducted, there were no apparent differences between the sexes.”
Lines 585-588: “Fifth, due to the small number of samples, human AH estradiol concentrations were not able to be statistically analyzed on the basis of sex. Furthermore, the biological sex of bovine and porcine AH was unknown although it was assumed to be male due to commonly accepted abattoir practices.”
Lines 771-773: “Written consent was collected from all individuals. Samples had been collected from male and female patients undergoing cataract or glaucoma surgery (Supplemental Table S6).”
Round 2
Reviewer 1 Report
The paper can be accepted